# From Good Expectancy to Good Feelings: How Optimism Is Related to Subjective Well-Being in Chinese Adults through the Pathway of Coping

**DOI:** 10.3390/bs14030165

**Published:** 2024-02-22

**Authors:** Yao Zheng, Yubo Hou, Zhiwen Dong

**Affiliations:** Beijing Key Laboratory of Behavior and Mental Health, School of Psychological and Cognitive Sciences, Peking University, Beijing 100871, China; yaz700@stu.pku.edu.cn (Y.Z.); 2001110672@stu.pku.edu.cn (Z.D.)

**Keywords:** optimism, subjective well-being, positive and negative coping, mediation analysis

## Abstract

Positive psychology has attracted increasing attention from many scholars worldwide. There is a considerable body of knowledge on the relationship between optimism and subjective well-being (SWB). However, their mediation mechanism has not been fully studied, and most of the current conclusions were formulated within the context of Western culture, with a limited number of empirical studies specifically targeting Chinese people. Based on the theories of self-regulation and stress coping, our research aimed to validate the association between optimism and SWB among Chinese adults and further investigate the mediating effect of positive and negative coping in this relationship. In Study 1, using a national dataset from the Chinese General Social Survey (*N* = 12,582), we captured the direct positive relationship between optimism and SWB. In Study 2, taking a cross-sectional study (*N* = 272), we found the mediating effect of positive and negative coping in the relationship between optimism and SWB according to correlation and regression analysis. In Study 3, taking a cross-lagged study (*N* = 343), we reverified the results of Study 2 and found negative coping no longer played a role as a mediator after accounting for the factors of social desirability and state anxiety in the analysis. These findings are worthwhile for paying attention to Chinese people’s optimistic traits and the pathways to improving their subjective well-being using different coping behaviors.

## 1. Introduction

With the development of positive psychology in recent decades across the world, a growing number of studies and researchers have shifted their attention from reducing psychiatric and psychological disorders in clinical patients to increasing the mental health and well-being of common people [1]. Among these studies, one continuously proposed argument was that genetics and personality had key impacts on our levels of subjective wellbeing (SWB) and human happiness [2,3,4,5]. The important individual characteristics that were associated with SWB and its subscales included the big five factors of extraversion, conscientiousness, and emotional stability [6,7,8]. Optimism, as an indicator of personal disposition, is generally conceptualized as an attitude toward or cognitive expectancy of positive outcomes [9]. It is believed that people with greater optimism tend to be more favorably adjusted to difficulties, associated with lower levels of anxiety and depression and better physical or psychological wellbeing. In contrast, less optimistic people always expect negative outcomes to happen and thus are more prone to experiencing anxiety, fearfulness, or depression in life [10] (pp. 402–407). It has also been examined that optimism is reliably correlated with improved physical or psychological health, enhanced effective coping skills, and even a higher quality of life and social relationships [11,12,13].

While the evidence from previous research has demonstrated a close relationship between optimism and well-being outcomes, our understanding of their mediating mechanism remains modest so far [12,14]. According to the stress and coping theory of Lazarus and Folkman (1984), which emphasized the importance of coping in the relationship between personality and psychological adjustment, it is reasonable to support the idea that optimism can be correlated with SWB according to a routine of various coping styles, such as self-control strategies, cognitive reappraisal, emotional regulation, and social supports [15,16,17]. However, most of the prior research has primarily focused on a particular type of coping without regarding it as a broad type of coping (i.e., positive vs. negative coping) in its analyses, and the conclusions were not consistent with each other [15,17]. Moreover, the majority of current research has only been conducted in W.E.I.R.D. (Western, Educated, Industrial, Rich, and Democratic) countries, particularly in the U.S. [18], which makes it worthwhile to further explore the relationships between optimism, coping, and SWB in some East Asian countries.

Therefore, to fill the literature gap and extend the findings in this area, the purposes of the current research would be to reverify the impact of optimism on SWB by collecting data from multiple sources in Chinese society, as well as to explore the mediating roles of positive and negative coping in the relationship between optimism and SWB, after accounting for the basic demographic and societal variables in general circumstances.

### 1.1. Optimism and SWB

Optimism is generally conceptualized as involving positive cognition and favorable attitudes toward the future [19]. Two predominant and congruent theories are commonly employed to elucidate the functions and impacts of optimism on outcomes. The first is Seligman’s [20] explanatory style model, which underscores the ways individuals explain past positive or negative events. Those who attribute positive events to internal and stable causes are considered optimists and are often characterized by their characteristics of persistence and hope and expectations of a better future [21]. The second model is the self-regulatory theory proposed by Carver and Scheier [22]. In this model, optimism involves a series of self-identification and assessment processes, motivating active engagement in daily life. When individuals perceive achievable goals and anticipate favorable results, they are more likely to employ increased effort and effective coping skills, even when facing difficulties or challenges [23]. For the sake of simplicity, this study focuses mainly on the trait-like nature of optimism in influencing its psychological outcome, SWB, in Chinese individuals, while it is worth noting that researchers have acknowledged the potential for optimism to exhibit variations over time [21,24].

As a specific domain of mental health and well-being, SWB represents individual evaluations of personal lives, encompassing cognitive and affective dimensions. It is commonly operationalized through measures such as life satisfaction (LS), representing one’s satisfaction or enjoyment in life; positive affect (PA), indicating the frequency of positive experiences; and negative affect (NA), reflecting the frequency of negative experiences in life [2,25]. While the tripartite structure of SWB is popular among researchers, the use of a single-item measurement for SWB has also been deemed appropriate and effective in certain studies [26]. Additionally, concepts such as quality of life and global or domain-specific life satisfaction (i.e., academic satisfaction) could also be employed as proxies for SWB in research [3,13,14,27]. Alternative methods for assessing SWB encompass physiological measures, memory and reaction time measures, and experience-sampling methods (ESM), aiming for a more comprehensive understanding of changes in feelings over time [2] (p. 35).

Based on a comprehensive review of the findings regarding the impact of trait optimism on SWB, the previous research encompasses a diverse array of contexts and participants [2]. These include clinical patients in medical settings, school students undergoing life transitions, and workers or entrepreneurs in organizational environments [11,27,28]. To date, two main interests of research have been conducted. The first type includes cross-sectional studies, revealing that lower optimism is associated with increased emotional distress and psychological disorders during challenging times [23,27]. Conversely, higher optimism correlates with greater life satisfaction and more positive emotions, coupled with reduced negative emotions [28,29,30]. The second type involves longitudinal studies, demonstrating that greater optimism not only is linked to fewer depression symptoms but also predicts a higher life satisfaction and quality of life at subsequent time points, even after accounting for the initial level of emotions [10,14,31]. Evidently, the second type of research provides more robust evidence supporting the beneficial effects of optimism on mental health and SWB compared to the first type. Overall, it is suggested that optimism, particularly trait optimism, maintains a positive association with SWB and its three components. Consequently, we propose the following hypothesis.

**Hypothesis** **1 (H1):**Optimism is significantly and positively associated with SWB, LS, and PA and negatively associated with NA.

### 1.2. Optimism, Coping, and SWB

While numerous empirical studies have demonstrated a close association between optimism and SWB in the West, the inner mechanism of this relationship is not fully understood [12,32]. Notably, there is a scarcity of studies conducted specifically with samples of Asian or Chinese populations in this field [16]. Considering the motivational component and goal-oriented behaviors associated with optimism, it is plausible to view actively coping with obstacles as an effective means for optimists to achieve SWB, along with other psychological adjustments [11,15,23]. In a meta-analysis focusing on optimism and coping relationships, a lot of evidence supported the association between optimism and different coping strategies, such as engagement vs. disengagement coping or problem-focused vs. emotion-focused coping [33]. In summary, optimism had a positive impact on approach (or positive) coping strategies like planning, behavior engagement, and support-seeking, whereas exhibiting a negative impact on avoidance (or negative) coping strategies, such as emotional suppression, behavior disengagement, and wishful thinking. Thus, we propose the following hypotheses.

**Hypothesis** **2 (H2):**Optimism is significantly and positively associated with positive coping.

**Hypothesis** **3 (H3):**Optimism is significantly and negatively associated with negative coping.

In line with stress and coping theory, individuals are inclined to experience less stress and more positive feelings when effectively coping with life stressors [34]. Recent research evidence also suggests that SWB can be influenced by different types of adaptive and maladaptive coping strategies [16,17]. Adaptive coping strategies, such as problem-focused coping and assistance pursuit coping, are associated with higher happiness and SWB, whereas maladaptive coping strategies, such as emotion-focused coping and low work control, are linked to a lower quality of life and SWB [17,35]. Consequently, we posit that positive and negative coping strategies may have opposing impacts on SWB. Specifically, positive coping strategies are expected to have favorable influences on SWB. In contrast, negative coping skills are anticipated to have unfavorable impacts on SWB. Therefore, we propose the following hypotheses.

**Hypothesis** **4 (H4):**Positive coping is significantly and positively associated with SWB, LS, and PA and negatively associated with NA.

**Hypothesis** **5 (H5):**Negative coping is significantly and negatively associated with SWB, LS, and PA and positively associated with NA.

Generally, individuals with optimism tend to possess self-confidence and expectations conducive to effective problem-solving and the development of a high-quality life over the long term [23]. Consequently, in theory, optimism should be associated with positive coping styles, leading to reduced experiences of stress and elevated levels of SWB. Consistent findings from previous studies indicated that optimism may be related to mental health and SWB according to various coping styles, such as increased social support, fighting spirit, humor, and task-oriented coping, particularly in the face of stressful situations and poor health [11,13,23]. However, the recent research has presented inconsistent findings, revealing the weak and insignificant indirect effects of optimism on SWB according to some types of coping strategies, such as primary and secondary self-control, in a sample of Japanese individuals [15] (pp. 41–42). These disparities may be attributed to either the mixed and intricate effects coping has on SWB or the distinctions in the sample characteristics between Eastern and Western populations [15]. Therefore, a revisit of the examination of optimism’s impact on SWB through coping strategies seems necessary. Building on the previous research and theories, we build a complete research model (see Figure 1) and propose the following hypotheses.

**Hypothesis** **6 (H6):**Optimism is significantly linked to SWB through positive coping.

**Hypothesis** **7 (H7):**Optimism is significantly linked to SWB through negative coping.

### 1.3. The Present Research

With the purpose of investigating the relationships between optimism, coping, and SWB in Chinese adults, three empirical studies were conducted to corroborate these hypotheses. In Study 1, we examined the direct effect of optimism on SWB by using national-wide big data. In Study 2, a cross-sectional study was conducted to explore the relationships between optimism, coping, and SWB, and the mediating effects of positive and negative coping were examined as well. In Study 3, the results found in Study 2 were reverified using a cross-lagged method design. All the data analyses in these studies were conducted using SPSS 27.0.

## 2. Study 1

Study 1 aimed to offer preliminary evidence supporting H1, which posits the positive relationship between optimism and SWB, after accounting for the control variables of gender, age, education level, and socioeconomic status (SES). The study utilized a large nationally representative dataset for its analysis.

### 2.1. The Data Source

The data for this study were sourced from the 2017 Chinese General Social Survey (CGSS 2017), a nationwide social survey project conducted by Renmin University of China (http://cgss.ruc.edu.cn/). The dataset consisted of responses from a total of 12,582 adult participants (≥18 years), covering all 31 provinces in China’s mainland. The sample was characterized by 52.80% female participants, with an average age of 51.01 (*SD* = 16.86), and 19.7% of participants holding a bachelor’s degree or above. The CGSS dataset has been utilized to explore the indicators of social and behavioral psychology in Chinese society [36,37,38]. This research adhered to the ethical guidelines set forth by the research ethics committee of the authors’ university, as well as subsequent studies.

### 2.2. Measures

Optimism: Three sentences, particularly capturing individual’s positive expectations about the future (i.e., “When things are uncertain, I usually hope for the best”; “I am optimistic about my future”; and “On balance, I expect more good things to happen to me”), based on the original version of the Life Orientation Test (LOT) developed by Scheier and Carver [39], were employed to assess the optimism in Chinese people, which was found to be effective in another recent body of research [37]. The respondents were required to rate each question on a 5-point scale, ranging from 1 = “very likely agree” to 5 = “very likely disagree”. Each item was inversely scored, and the average of all items was computed as the final score in analysis, with a higher score indicating a higher level of optimism in the long term. The internal consistency of the scale in this study was 0.66, which showed an acceptable level of reliability as one previous research work undertaken by Lai and Yue [40].

Subjective well-being (SWB): SWB was assessed using a single-item measure of the 36th question in CGSS 2017, referring to some of the latest research in Chinese samples [37,38]. The question was “In general, how much can you feel happy in your life?”. The specific answers included 1 = “very unhappy”, 2 = “unhappy”, 3 = “not possible to say happy or unhappy”, 4 = “happy”, and 5 = “very happy”. A higher score on this item indicated a higher level of SWB in the long term.

Control Variables: The control variables in this study included the major demographic variables, such as age, gender, education level, and a socioeconomic variable of SES, which were believed to have close relationships with optimism, as well as well-being outcomes, in prior research [16,18,23,27,31].

### 2.3. Results

The descriptive statistics and correlations for the variables are presented in Table 1, which showed that optimism was significantly and positively associated with SWB (r = 0.31, *p* < 0.001). Additionally, a regression analysis was conducted, with SWB regressed on gender, age, education level, SES, and optimism (see Table 2). The findings indicated that optimism had a significant and positive association with SWB (ß = 0.25, *p* < 0.001), even after controlling for the major demographic variables. The observed pattern of results remained consistent when no control variables were included, thus approving H1.

## 3. Study 2

In Study 1, we established preliminary evidence indicating a significant positive association between optimism and SWB. However, the assessment of SWB was limited to a single-item measure. In Study 2, we sought to replicate the findings of Study 1 using a more comprehensive measure of the tripartite structure of SWB and a new Chinese adult sample in Beijing colleges. Additionally, we investigated the mediating effects of positive and negative coping in the relationship between optimism and SWB after accounting for the key demographic and socioeconomic variables.

### 3.1. Participants and Procedure

A total of 300 college student respondents were recruited using the online bulletin board systems (BBS) of three colleges in the Beijing area, China. After excluding scores indicating inattentiveness (*N* = 23), participants that took too long or short a time to complete the questionnaires, and individuals below the age of 18 (*N* = 5), a total of 272 participants remained in the study. The final sample comprised 42.6% female participants, with an average age of 23.22 (*SD* = 3.63), and more than 90% of the participants had attained a bachelor’s degree or higher.

All participants were invited to complete a battery of questionnaires covering optimism, Chinese coping style, subjective well-being, and their basic demographic information. After reading a consent letter and completing all the survey items, participants were eligible to receive a cash payment contingent upon the acceptance of their scores at the end of the survey.

### 3.2. Measures

Optimism: The Chinese Revised Life Orientation Test (CLOT-R), developed by Lai and Yue [39], was utilized to measure participants’ optimism, which was regarded as a reliable and efficient instrument in the Chinese samples [37,38]. This scale, originally developed by Scheier and Carver [39], was later revised by Scheier et al. [9] to include 10 self-report items. Out of these items, three were utilized to measure dispositional optimism (i.e., ‘‘I’m always optimistic about my future’’), and three items were reverse-coded (i.e., “I rarely expect good things to happen to me”). The remaining four items served as fillers without analysis. Respondents rated each item on a 5-point scale, ranging from 1 = “strongly disagree” to 5 “strongly agree” (Cronbach’s *α* = 0.68), with a higher score representing a higher level of optimism.

Coping Strategies: The Simplified Coping Style Questionnaire (SCSQ) developed by Xie [41] was employed to measure the positive and negative coping styles of participants, which was found to be a valid measure in Chinese people [42]. Adapted from the Ways of Coping Questionnaire (WCQ) by Folkman and Lazarus [43], this 20-item instrument included the dimensions of positive coping (PC, 12 items, Cronbach’s *α* = 0.80) and negative coping (NC, 8 items, Cronbach’s *α* = 0.76). Positive coping involves proactive strategies such as problem-solving, seeking help, and cognitive reappraisal, while negative coping pertains to inactive strategies, like denial, avoidance, imagination, and acceptance. Each item was rated on a 4-point scale, ranging from 1 = “never” to 4 = “always”, with a higher score indicating a higher frequency of applying such types of coping strategies.

Subjective Well-being: The Satisfaction with Life Scale (SWLS) developed by Diener [44] and the Positive and Negative Affect Schedule (PANAS) from Watson et al. [45] were applied to measure SWB and its three components, as mentioned above. These scales have shown good reliability and validity in previous studies and have been widely used across different cultures [15,31,35]. The items from them were rated on a 5-point scale (SWLS: from 1 = “very unlikely fit” to 5 = “very likely fit”; PANAS: from 1 = “never” to 5 = “always”). Higher scores indicated higher levels of life satisfaction (LS, 5 items, Cronbach’s *α* = 0.86), positive affect (PA, 6 items, Cronbach’s *α* = 0.87), and negative affect (NA, 6 items, Cronbach’s *α* = 0.88). The final SWB score was calculated by adding the average scores of LS and PA and then subtracting the average score of NA [25].

### 3.3. Results

#### 3.3.1. Testing for Common Method Bias

Given that the participants reported all data simultaneously in this study, an initial common method bias test was conducted. Harman’s single-factor test was employed to examine the potential common method bias [46]. All items were included in a single factor, and the unrotated factor revealed that the cumulative variance explained by nine factors with eigenvalues greater than 1 was 59.79%. The factor with the largest eigenvalue explained 24.37% of the variance, which was less than the threshold of 40%, indicating that the impact of common method bias on this study was not of substantial consequence.

#### 3.3.2. Testing for the Total Effect

The results from the correlation analysis (see Table 3) showed that optimism was statistically significantly associated with SWB (r = 0.63, *p* < 0.001), as well as the constituents of LS (r = 0.42, *p* < 0.001), PA (r = 0.54, *p* < 0.001), and NA (r = −0.52, *p* < 0.001). Regression analysis was conducted to examine the direct impact of optimism on SWB and its three components. The results in Table 4, controlling for age, gender, education level, and SES, revealed a positive relationship between optimism and SWB (ß = 0.59, *p* < 0.001), including positive relationships with LS (ß = 0.38, *p* < 0.001) and PA (ß = 0.51, *p* < 0.001) and a negative relationship with NA (ß = −0.50, *p* = 0.001), thus confirming H1. These findings were consistent even after removing the control variables.

#### 3.3.3. Testing for a Mediating Effect

The multiple regression analysis (see Table 4) showed that, after controlling for gender, age, education level, and SES, optimism was positively related to positive coping (ß = 0.51, *p* < 0.001) and negatively related to negative coping (ß = −0.36, *p* < 0.001), supporting H2 and H3. Additionally, positive coping was significantly related to SWB (ß = 0.39, *p* < 0.001) and its three components of LS (ß = 0.37, *p* < 0.001), PA (ß = 0.36, *p* < 0.001), and NA (ß = −0.16, *p* < 0.05) while accounting for the effect of optimism, thus confirming H4. To the contrary, controlling optimism, negative coping was no longer significantly correlated with SWB (ß = −0.03, *p* = 0.56), LA (ß = 0.11, *p* = 0.05) and PA (ß = 0.05, *p* = 0.34), with the exception of a positive relationship with NA (ß = 0.27, *p* < 0.001), thus resulting in H5 being unconfirmed.

“Model 4” in PROCESS macro v4.0 [47] was employed to test the indirect effects based on 5000 bootstrap samples [48]. When covariates were entered, significant indirect effects of optimism on SWB through positive coping (*effect size* = 0.21, 95%CI = [0.16, 0.28]) and negative coping (*effect size* = 0.05, 95%CI = [0.01, 0.08]) were observed, respectively (see Figure 2). Thus, H6 and H7 were confirmed in this study.

## 4. Study 3

In Study 2, we found a positive effect of optimism on SWB and the mediating effects of positive and negative coping in the relationship between optimism and SWB by undertaking a cross-sectional study. In Study 3, a cross-lagged study was designed by collecting data across three different time points. According to the previous research, it is argued that optimism could be influenced by participants’ initial level of emotion [11] and societal anxiety, which was common in most collectivistic countries [14]. There is also a possible positive response bias in measuring social behaviors and psychology [5,7]. Therefore, the results in this study were analyzed excluding the impacts of social desirability and state anxiety.

### 4.1. Participants and Procedure

The study employed three-wave questionnaires with one- to two-week intervals using the platform Credamo (https://www.credamo.com) on 10 July 2023. At Time 1, 450 adult participants completed scales measuring their optimism, state anxiety, social desirability, and basic demographics. At Time 2, 381 respondents remained to answer the two subscales of positive and negative coping styles. At Time 3, questionnaires assessing LS, PA, and NA were assigned simultaneously, resulting in a final sample of 343 participants (61.2% female, *M*_age_ = 30.97, *SD*_age_ = 5.67). The attrition rate for this study, ranging from 15% to 10% between Time 1 and Time 3, adheres to the established research standards [13,14]. Participants had the option to withdraw from the study at any point.

### 4.2. Measures

Optimism: The same scale used in Study 2 was employed to measure participants’ optimism. The internal consistency reliability of the scale in this study was 0.75.

Coping Strategies: The same scale from Study 2 was utilized to measure the two dimensions of Chinese coping styles. The internal consistency reliability of the subscales in this study was 0.55 for PC and 0.70 for NC.

Subjective Well-being: The same measurements from Study 2 were employed to assess SWB and its components. The internal consistency reliability in this study was 0.80 for LS, 0.86 for PA, and 0.69 for NA.

State Anxiety: The Self-Rating Anxiety Scale (SAS), developed by Zung [49], served as the instrument for assessing participants’ state anxiety and had good psychometric properties in prior studies [49]. The self-report scale includes 20 items covering various anxiety symptoms, and participants rated their current experiences over the last week on a 4-point Likert scale, from 1 = “a little of the time” to 4 = “all of the time”, with a higher score indicating greater anxiety. The internal consistency reliability of the scale in this study was 0.79.

Social Desirability: The 13-item short version of the Marlowe Crowne Social Desirability (MCSD) scale, based on the original version from Reynolds [50], was often used to measure possible social desirability bias in the Chinese samples [51]. Each item was rated on a Y/N answer sheet, and participants judged whether a statement was true or false for themselves. In the data analysis, 11 items were inversely coded, and a higher score indicated a higher level of social desirability. The internal consistency reliability of the scale in this study was 0.88.

### 4.3. Results

#### 4.3.1. Testing of Common Method Bias

Although the data were collected across multiple time points, all data were self-reported by the participants, and the risk of common method bias still existed. Therefore, we used Harman’s single-factor test to conduct a common method bias test [46] and put all items into the same factor in the factor analysis. The results indicated that 65.67% of the cumulative variance was explained by the top 20 factors, with eigenvalues greater than 1, and the largest eigenvalue could explain 19.14% of the total variance, which was lower than the boundary value of 40%. This outcome suggested that the common method bias in this study did not have a significant influence on the study results.

#### 4.3.2. Testing for the Total Effect

As shown in Table 5, participants’ optimism had a significant positive correlation with SWB (r = 0.72, *p* < 0.001) and its components of LS (r = 0.66, *p* < 0.001) and PA (r = 0.65, *p* < 0.001), while it had a negative correlation with NA (r = −0.49, *p* < 0.001). A regression analysis was performed to examine the impact of optimism on people’s SWB and its components as well. The results revealed that optimism was significantly and positively related with SWB (ß = 0.46, *p* < 0.001) and the components of LS (ß = 0.48, *p* < 0.001) and PA (ß = 0.44, *p* < 0.001). In contrast, optimism showed a negative relationship with NA (ß = −0.17, *p* = 0.001) in SWB, even after controlling for the demographic variables and the initial levels of state anxiety and social desirability. These findings were consistent in this study, even after removing these control variables.

#### 4.3.3. Testing for a Mediating Effect

The results of the multiple regression analysis, presented in Table 6, demonstrated that after adjusting for gender, age, education level, SES, state anxiety, and social desirability, optimism still exhibited a significant positive association with positive coping (ß = 0.21, *p* < 0.001) and a negative association with negative coping (ß = −0.25, *p* < 0.001). Hence, H2 and H3 were supported in this study. Subsequently, after accounting for the level of optimism, positive coping was still significantly associated with SWB (ß = 0.16, *p* < 0.001) and its component of PA (ß = 0.27, *p* < 0.001), thus almost supporting H4. However, after controlling for optimism, negative coping was no longer significantly related to SWB (ß = −0.07, *p* = 0.06) and its components of LA (ß = −0.08, *p* = 0.07) and PA (ß = −0.02, *p* = 0.63). Therefore, H5 was not confirmed in this study.

Subsequently, “model 4” in PROCESS macro v4.0 [47] was employed to test the indirect effects based on 5000 bootstrap samples [48]. With age, gender, education level, SES, state anxiety, and social desirability entered as covariates, the results demonstrated that there was a significant indirect effect of optimism on SWB via the routine of positive coping (*effect size* = 0.04, 95%CI = [0.01, 0.07]). However, optimism was no longer significantly correlated with SWB through the pathway of negative coping (*effect size* = 0.02, 95%CI = [−0.001, 0.04]) after accounting for the effect of social desirability (see Figure 3). Therefore, this study provided support for H6 but not for H7.

## 5. Discussion

Grounded in the perspective of personal character strengths and positive cognition, our research explored the relationships between optimism, coping, and SWB, as well as the mediating mechanism via the pathway of Chinese coping behaviors, across three studies.

First, consistent with past conclusions, the present research underscored a significant correlation between optimism and SWB, encompassing its three constituent elements: LS, PA, and NA. Specifically, individuals with higher optimism exhibited greater life satisfaction and experienced heightened positive emotions and reduced negative emotions, thereby perceiving an elevated overall SWB in life [14,27,31]. Similar findings showed that positive expectancies could predict lower depression, even after controlling for the baseline level [29]. In addition, according to Long’s research [15], although Americans showed more optimism compared to Japanese adults, the relationship between optimism and SWB remained positive for both groups of people. Importantly, such a conclusion was also confirmed for Chinese adults, which suggested that optimism could exert an impact on SWB under different cultures.

Second, optimism was positively associated with positive coping (i.e., problem-solving strategy) and negatively associated with negative coping (i.e., avoidance), which was consistent with most of the previous research [13,23,30]. For example, Brissette et al.’s [11] research indicated that students with higher optimism tended to use more active coping and use denial and disengagement strategies less during their first semester of school. Furthermore, a meta-analysis by Nes and Segerstrom [33] concluded that optimistic expectations, instead of disengaging and giving up, could lead to more active engagement and continued striving to achieving one’s goals. Such a contention was supported by the empirical studies in this review.

Third, we found out that positive coping retained a positive relationship with SWB in two studies, while negative coping was no longer significantly correlated with SWB after accounting for the main effect of optimism on SWB. These findings were partially supported by the previous research [11,13,35], which showed that frequent use of active and positive coping could result in smaller increases in depression and stress and higher levels of satisfaction and happiness in life. In contrast, negative coping styles, such as behavior disengagement and denial, were correlated with less psychological adjustment in general [11,16]. This contradiction to our results may be attributed to the wide categories of negative coping that were measured in different research works. Specific coping styles may have different impacts on individuals’ mental health [17,52]. For instance, Fischer et al. [17] proposed that negative coping strategies, like “avoidance”, could function quite differently and exert either a positive or negative effect on outcomes depending on the specific situation (p. 3828). Whether other kinds of negative coping could show an unstable and context-dependent association with SWB needs further exploration in the future.

Lastly, we identified the mediating role of positive coping in the relationship between optimism and SWB. Such a result was in line with the prior research [10,11,15]. For example, a longitudinal survey by Kaida [31] explored the dynamic changes in the associations between optimism, pro-environmental behaviors (PEBs), and SWB and found optimism could be indirectly correlated with current and future SWB according to the active coping mechanism of PEBs. However, contrary to the hypothesis, negative coping could not be maintained as a mediator in the impact of optimism on SWB after accounting for the effects of state anxiety and social desirability in the analysis. In addition to the mentioned reasons for the various effects of negative coping and sample variances across cultures [15,17], one latest research highlighted that adaptive coping may exert a more critical impact on psychological well-being compared to less use of maladaptive coping [52]. Furthermore, in Liao’s research [42], negative coping, such as emotional venting and attention-shifting coping, was perceived as a mixed and complex coping style for Chinese people and demonstrated a weak positive correlation with mental health, which could help provide another perspective for explaining the current research.

### 5.1. Theoretical and Practical Implications

This study offered theoretical and practical implications to research in the area of positive and personality psychology, deepening our understanding of the relationship between optimism and SWB in Chinese people. First, we highlighted the importance of optimism in fostering happiness and SWB outside the scope of Westerners. Despite the prior work devoting a lot of time to investigating the influence of optimism on reducing psychological illness or symptoms [10], few bodies of research have explored why and how optimism could improve people’s mental health or well-being, especially when considering that well-being is not simply the opposite of illness or psychopathology [2,11]. Moreover, this study revealed that, in terms of coping mechanisms, the impact of optimism on SWB could persistently be achieved using positive coping. In comparison, negative coping played a role as a weak mediator in the relationship between the optimism and SWB of Chinese people. Thus, more research is required to classify whether some negative coping strategies have more complex and varying impacts on SWB depending on the specific circumstances [17].

Second, as a supplement to the previous research confined to the Western populations [15,16], our research provided reassurance that there was a significant and positive effect of optimism on SWB around the world, which provided support for policymakers, therapists, and organizational managers to make greater efforts to increase individuals’ happiness and well-being in practice. Proactive measures and interventions aimed at cultivating optimism may involve the development of future-oriented goals and visualizing our best possible selves [53]. Moreover, given that people with different cultural backgrounds could differ in their specific motives for and values in SWB [2,3], clinical psychologists and educational practitioners should pay more attention to improving Chinese people’s active coping skills, such as humor and cognitive reappraisal [16], rather than simply inhibiting the negative coping strategies people are engaged in, such as emotional venting and diverting attention from problems [52]. It has been proven in this research that positive coping strategies, rather than negative coping styles, are more important to fostering Chinese happiness and SWB.

### 5.2. Limitations and Future Directions

There are still some limitations to the current research. First, the fact that all the data were collected online made it indecisive whether the relationships between optimism, coping, and SWB in this study would persist when conducting research in more realistic field conditions. Second, our research mainly focused on a group of Chinese adults aged 18 and above. We cannot ensure whether the results in the present study are applicable to a younger and wider group of people, including adolescents in middle or high school and people located in other Asian countries (i.e., India, Japan, and Korea). Third, although major demographic and socioeconomic variables were controlled in the present study, other important factors, such as occupation, family income, regional differences, and current mood, should also be considered in future research [16]. Fourth, due to the low internal reliability of some of the scales used in the analysis, the correlations between variables may be underestimated. Therefore, other alternative and more culturally relevant measures of optimism and coping should be considered in the future.

Moreover, we solely examined the predictive power of optimism for SWB and its three components according to the mediator of coping behaviors. Comparable concepts, such as psychological well-being and hedonic and eudemonic well-being, were not thoroughly investigated in the present research [16,29]. Future studies could focus on assessing the impacts of optimism and coping, either respectively or simultaneously, on influencing individuals’ autonomy, positive relations, purpose in life, and so forth [5].

Finally, based on the self-reporting nature of all the current measures, other measurement methods, especially experimental and objective-based approaches to assessing the concepts of optimism, coping, and SWB, should be employed in the future to further explore the causal relationships among variables and better classify the benefits optimism could bring to the promotion of coping styles and SWB in diverse situations and environments.

## 6. Conclusions

Our study primarily sought to explore the relationships among the concepts of optimism, coping, and SWB in Chinese adults, aiming to broaden our understanding of the impact of trait optimism on human happiness and psychological adjustment from the perspective of positive cognitive constructs. Based on the theories of self-regulation and stress coping, the results from three empirical studies revealed that optimism had a significant and stable impact on positive and negative coping styles, as well as SWB and its components. Specifically, optimism could be consistently and positively associated with positive coping, SWB, and the domains of LS and PA, whereas it was negatively correlated with negative coping and the subdomain of NA in SWB. Furthermore, the findings of the current research suggested that there was only a weak negative association between negative coping and SWB, especially when excluding the main effect of optimism on SWB. Again, positive coping, instead of negative coping, plays a mediating role in the relationship between optimism and SWB after accounting for the effects of social desirability and state anxiety on the participants. In this regard, SWB and its components would be largely influenced by optimism according to a routine of positive coping, which tends to be more independent of changes in the environment and situations compared to the varying and context-dependent effects of negative coping on well-being outcomes.

## Figures and Tables

**Figure 1 behavsci-14-00165-f001:**
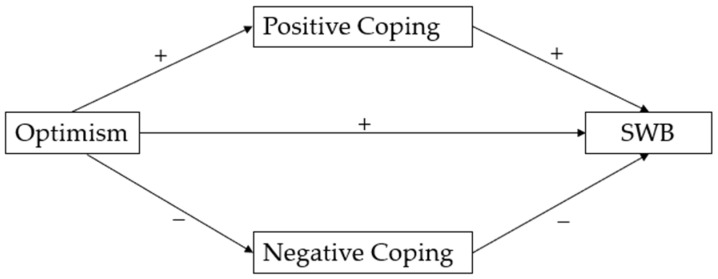
The hypothesized conceptual model.

**Figure 2 behavsci-14-00165-f002:**
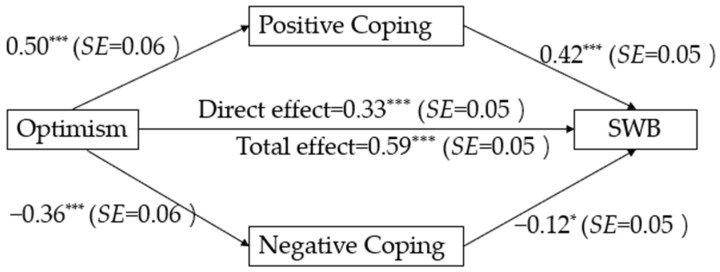
Regression results of the full model in Study 2 (standardized coefficients) * *p* < 0.05, *** *p* < 0.001.

**Figure 3 behavsci-14-00165-f003:**
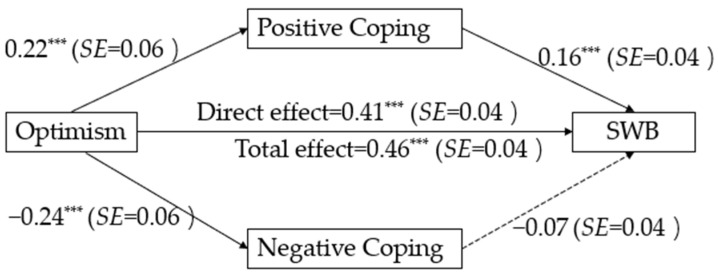
Regression results of the full model in Study 3 (standardized coefficients) *** *p* < 0.001.

**Table 1 behavsci-14-00165-t001:** The description and correlation of main variables in Study 1.

	*M*	*SD*	1	2	3	4	5
1. Gender	—	—	1				
2. Age	51.01	16.86	−0.01	1			
3. Edu.	5.17	3.28	−0.10 ***	−0.47 ***	1		
4. SES	2.22	0.87	0.02 **	−0.05 ***	0.24 ***	1	
5. Opti.	3.84	0.59	0.02	−0.07 ***	0.14 ***	0.20 ***	1
6. SWB	3.86	0.85	0.03 ***	0.01	0.13 ***	0.29 ***	0.31 ***

Note: *N* = 12,582. *M* = mean. *SD* = standard deviation. Edu. = education level. SES = socioeconomic status. Opti. = optimism. SWB = subjective well-being. Gender was recorded as a dummy variable (0 = male, 1 = female). ** *p* < 0.01, *** *p* < 0.001.

**Table 2 behavsci-14-00165-t002:** Results of regression analyses in Study 1.

Variable	*DV*: General SWB			
	ß	*SE*	ß	*SE*
Gender	0.04 ***	0.01	0.05 **	0.01
Age	0.08 ***	0.01	0.06 ***	0.02
Edu.	0.11 ***	0.01	0.08 ***	0.02
SES	0.27 ***	0.01	0.24 ***	0.02
Opti.			0.25 ***	0.01
*R* ^2^	0.10		0.16	

Note: ß = standardized coefficient. *SE* = standard error. Edu. = education level. SES = socioeconomic status. Opti. = optimism. SWB = subjective well-being. ** *p* < 0.01, *** *p* < 0.001.

**Table 3 behavsci-14-00165-t003:** The description and correlation of main variables in Study 2.

	*M*	*SD*	1	2	3	4	5	6	7	8	9	10
1. Gender	—	—	1									
2. Age	23.22	3.61	−0.03	1								
3. Edu.	4.87	0.68	0.01	0.16 **	1							
4. SES	5.69	1.75	−0.02	−0.00	0.08	1						
5. Opti.	3.51	0.66	0.08	0.04	0.16 **	0.17 **	1					
6. PC	3.01	0.44	−0.04	0.03	0.16 *	0.22 ***	0.53 ***	1				
7. NC	2.53	0.55	−0.14 *	0.01	−0.08	0.10	−0.35 ***	0.02	1			
8. LS	3.64	0.85	−0.04	−0.00	0.12 *	0.33 ***	0.42 ***	0.52 ***	−0.01	1		
9. PA	2.84	0.65	−0.05	−0.03	0.12 *	0.28 ***	0.54 ***	0.57 **	−0.11	0.68 ***	1	
10. NA	1.76	0.64	−0.14 *	−0.01	−0.25 ***	−0.12	−0.52 ***	−0.38 ***	0.44 ***	−0.21 ***	−0.22 ***	1
11. SWB	4.72	1.65	0.02	−0.01	0.21 ***	0.28 ***	0.63 ***	0.64 ***	−0.22 ***	0.86 ***	0.83 ***	−0.56 ***

Note: *N* = 272. Edu. = education level. SES = socioeconomic status. Opti. = optimism. PC = positive coping style. NC = negative coping style. LS = life satisfaction. PA = positive affect. NA = negative affect. SWB = subjective well-being. Gender was recorded as a dummy variable (0 = male, 1 = female). * *p* < 0.05, ** *p* < 0.01, *** *p* < 0.001.

**Table 4 behavsci-14-00165-t004:** Results of regression analyses in Study 2.

Variable	PC (M_1_)		SWB		LS		PA		NA	
	ß	*SE*	ß	*SE*	ß	*SE*	ß	*SE*	ß	*SE*
Age	−0.00	0.01	−0.05	0.02	−0.03	0.01	−0.06	0.01	0.03	0.01
Gender	−0.08	0.05	0.01	0.14	−0.03	0.09	−0.06	0.06	−0.11	0.07
Edu.	0.07	0.03	0.08	0.10	0.02	0.06	0.01	0.05	−0.17	0.05
SES	0.12 *	0.01	0.12 **	0.04	0.21 ***	0.03	0.14 **	0.02	0.11 *	0.02
Opti.	0.51 ***	0.04	0.39 ***	0.12	0.19 **	0.08	0.33 ***	0.05	−0.42 ***	0.06
PC (M_1_)			0.39 ***	0.18	0.37 ***	0.11	0.36 ***	0.08	−0.16 *	0.09
*R* ^2^	0.31		0.55		0.34		0.43		0.34	
**Variable**	**NC (M2)**		**SWB**		**LS**		**PA**		**NA**	
	**ß**	* **SE** *	**ß**	* **SE** *	**ß**	* **SE** *	**ß**	* **SE** *	**ß**	* **SE** *
Age	0.03	0.01	−0.05	0.02	−0.03	0.01	−0.06	0.01	0.03	0.01
Gender	−0.11	0.06	−0.03	0.15	−0.05	0.09	−0.08	0.07	−0.07	0.06
Edu.	−0.04	0.05	0.11 *	0.11	0.05	0.07	0.04	0.05	−0.17 **	0.05
SES	0.16 **	0.02	0.18 ***	0.04	0.24 ***	0.03	0.18 **	0.02	0.04	0.02
Opti.	−0.36 ***	0.05	0.58 ***	0.13	0.42 ***	0.08	0.53 ***	0.05	−0.40 ***	0.05
NC (M_2_)			−0.03	0.15	0.11	0.09	0.05	0.06	0.27 ***	0.06
*R* ^2^	0.16		0.45		0.26		0.34		0.38	

Note: ß = standardized coefficient. *SE* = standard error. Edu. = education level. SES = socioeconomic status. Opti. = optimism. PC = positive coping style. NC = negative coping style. SWB = subjective well-being. LS = life satisfaction. PA = positive affect. NA = negative affect. Gender was recorded as a dummy variable (0 = male, 1 = female). * *p* < 0.05, ** *p* < 0.01, *** *p* < 0.001.

**Table 5 behavsci-14-00165-t005:** The description and correlation of main variables in Study 3.

	*M*	*SD*	1	2	3	4	5	6	7	8	9	10	11	12
1.Gender	—	—	1											
2. Age	30.97	5.67	−0.09	1										
3. Edu.	5.12	0.550	0.18 **	−0.06	1									
4. SES	5.74	1.43	0.09	0.09	0.22 ***	1								
5. SAS	1.46	0.27	0.07	−0.11 *	−0.13 *	−0.21 ***	1							
6. SD	9.51	3.39	0.00	−0.06	0.18 ***	0.25 ***	−0.27 ***	1						
7. Opti.	4.23	0.58	−0.02	0.13 *	0.21 ***	0.27 ***	−0.55 ***	0.43 ***	1					
8. PC	3.25	0.30	−0.01	0.14 **	0.13 *	0.15 **	−0.28 ***	0.32 ***	0.38 ***	1				
9. NC	2.08	0.47	−0.01	−0.13 *	−0.03	−0.03	0.12 *	−0.26 ***	−0.26 ***	−0.01	1			
10. LS	3.19	0.57	−0.04	0.10	0.18 **	0.34 ***	−0.46 ***	0.45 ***	0.66 ***	0.34 ***	−0.24 ***	1		
11. PA	3.21	0.55	−0.01	0.15	0.20 ***	0.31 ***	−0.50 ***	0.49 ***	0.65 ***	0.52 ***	−0.16 **	0.69 ***	1	
12. NA	1.28	0.30	0.05	−0.11 *	−0.08	−0.10	0.65 ***	−0.29 ***	−0.49 ***	−0.28 ***	0.20 ***	−0.52 ***	−0.51 ***	1
13. SWB	5.11	1.22	−0.04	0.10	0.19 ***	0.32 ***	−0.60 ***	0.50 ***	0.72 ***	0.46 ***	−0.24 ***	0.90 ***	0.90 ***	−0.72 ***

Note: *N* = 343. Edu. = education level. SES = socioeconomic status. SAS = state anxiety. SD = social desirability. Opti. = optimism. PC = positive coping style. NC = negative coping style. LS = life satisfaction. PA = positive affect. NA = negative affect. SWB = subjective well-being. Gender was recorded as a dummy variable (0 = male, 1 = female). * *p* < 0.05, ** *p* < 0.01, *** *p* < 0.001.

**Table 6 behavsci-14-00165-t006:** Results of regression analyses in Study 3.

Variable	PC (M_1_)		SWB		LS		PA		NA	
	ß	*SE*	ß	*SE*	ß	*SE*	ß	*SE*	ß	*SE*
Age	0.06	0.00	−0.02	0.01	0.00	0.00	−0.05	0.00	−0.03	0.00
Gender	−0.04	0.03	−0.01	0.09	−0.04	0.05	0.01	0.04	−0.01	0.033
Edu.	0.04	0.03	0.00	0.08	0.01	0.04	0.01	0.04	0.02	0.02
SES	0.02	0.01	0.08 *	0.03	0.14 **	0.02	0.08 *	0.02	0.09 *	0.01
SAS	−0.07	0.06	−0.26 ***	0.18	−0.11 *	0.10	−0.16 ***	0.09	0.55 ***	0.05
S.D.	0.25 ***	0.01	0.17 ***	0.02	0.16 **	0.01	0.17 ***	0.01	−0.08	0.00
Opti.	0.22 ***	0.03	0.43 ***	0.09	0.47 ***	0.05	0.38 ***	0.05	−0.16 **	0.03
PC (M_1_)			0.15 ***	0.16	0.05	0.09	0.26 ***	0.08	−0.05	0.05
*R* ^2^	0.21		0.64		0.50		0.57		0.47	
**Variable**	**NC (M_2_)**		**SWB**		**LS**		**PA**		**NA**	
	**ß**	** *SE* **	**ß**	** *SE* **	**ß**	** *SE* **	**ß**	** *SE* **	**ß**	** *SE* **
Age	−0.08	0.00	−0.01	0.03	−0.00	0.04	−0.04	0.04	−0.02	0.00
Gender	0.00	0.05	−0.01	0.03	−0.04	0.04	0.01	0.04	−0.01	0.03
Edu.	0.05	0.05	0.01	0.04	0.02	0.04	0.02	0.04	0.02	0.02
SES	0.07	0.02	0.08 *	0.04	0.14 **	0.04	0.08	0.04	0.08	0.01
SAS	−0.04	0.11	−0.27 ***	0.04	−0.12 *	0.05	−0.18 ***	0.05	0.55 ***	0.05
SD	−0.13 *	0.01	0.20 ***	0.04	0.16 ***	0.04	0.23 ***	0.04	−0.08	0.00
Opti.	−0.25 ***	0.05	0.45 ***	0.04	0.47 ***	0.05	0.44 ***	0.05	−0.15 **	0.03
NC (M_2_)			−0.04	0.04	−0.07	0.04	0.02	0.04	0.08	0.03
*R* ^2^	0.10		0.62		0.50		0.51		0.46	

Note: ß = standardized coefficient. *SE* = standard error. Edu. = education level. SES = socioeconomic status. SAS = state anxiety. SD = social desirability. Opti. = optimism. PC = positive coping style. NC = negative coping style. SWB = subjective well-being. LS = life satisfaction. PA = positive affect. NA = negative affect. Gender was recorded as a dummy variable (0 = male, 1 = female). * *p* < 0.05, ** *p* < 0.01, *** *p* < 0.001.

## Data Availability

Data are available upon request.

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
