# Peer review of "From Good Expectancy to Good Feelings: How Optimism Is Related to Subjective Well-Being in Chinese Adults through the Pathway of Coping"

_behavsci, 2024, doi:10.3390/bs14030165_

Round 1
Reviewer 1 Report
Comments and Suggestions for Authors
This study aimed to examine and confirm the connection between optimism and subjective well-being in a Chinese population while also exploring the potential mediating effect of positive and negative coping in this association. The authors have produced a paper that is both well-structured and well-written. However, I would like to provide some feedback and suggestions to improve the clarity and comprehensiveness of the paper.
The introduction section includes multiple paragraphs that are excessively detailed, which can potentially hinder the reader's understanding of the text's progression. As a result, it would be advantageous to simplify the material and offer a brief but comprehensive summary of the research.
The abstract's second line mentions the authors' acknowledgment of a significant amount of knowledge regarding the connection between optimism and subjective well-being (SWB). However, they also claim that the topic has not been fully explored, although there are numerous research projects on the matter. It is important to address the confusion and understand what sets this research apart from others beyond just the context.
In addition, the authors' use of various phrases suggests that they are exploring the relationship between variables. However, the authors have already addressed this hypothesis in previous research, indicating that they are working on a topic that has already been investigated. As a result, this paragraph may reduce the study's significance, suggesting that the authors' research may not provide new insights but rather reiterate existing knowledge. e.g., * Low levels of Optimism associated with More distress *High level of Optimism associated with Greater life satisfaction and more positive emotions, and lower depression. * Positive effect of optimism on SWB. And then the hypothesis is that: "Optimism is significantly and positively associated with SWB"
It is worth mentioning that the hypothesis lacks information regarding its directionality, i.e., whether it is bidirectional or unidirectional. This ambiguity needs to be addressed for all the hypotheses.
Additionally, various sources, including "Psychometric Theory" by Jum C. Nunnally and Ira H. Bernstein, suggest that a Cronbach's alpha of 0.7 is the minimum threshold for acceptable reliability in psychological research. The measurement used in the study has a Cronbach's alpha of 0.66, which is slightly below the commonly accepted standard. This indicates a reliability issue that raises concern. In addition, The internal consistency for the Positive Coping (PC) subscale was relatively low (Cronbach's alpha = 0.55). This could affect the reliability of the findings related to coping strategies. It is recommended that the authors address this in the discussion of their results, possibly considering alternative measures or an expanded scale for future research.
Moreover, there is no consideration in the introduction and discussion section regarding the role of demographic variables in the model, but it has been reported in the results section. It would be better to consider these variables to discuss the results. Before the Results' section, there was no mention of the role of age, gender, SES, and education level. Also, these should be mentioned in the aim of the study if they were to be measured!
The average age of participants in Study 1 was 51, whereas Study 2 included college students. As a result, there were significant age differences between the samples. Furthermore, it seems that there is no definite information provided about the criteria for inclusion or exclusion in all the studies.
The authors utilized the Subscale of the Chinese Revised Life Orientation Test (CLOT-R), which was developed by Lai et al. [39], in order to assess the participants' trait optimism. It is important to provide a reason why a specialized measurement tool catering specifically to the evaluation of optimism is not being used.
The study found a significant positive association between optimism and positive coping and a negative association with negative coping. While this supports hypotheses H2 and H3, hypothesis H5 (relating to the association between negative coping and SWB) was not confirmed. The manuscript would benefit from a more detailed discussion of these results, especially the non-confirmation of H5. How do these findings align with existing literature, and what implications might they have for future research in this area?
The conclusion section could be strengthened by providing clear implications of the findings for both theory and practice. Additionally, the authors should suggest specific areas for future research, especially considering the non-confirmed hypothesis and the lower reliability of the measurements.
Overall, the authors have produced an insightful paper that sheds light on an important topic. By implementing these suggestions, the paper can be further refined and contribute to the existing body of research in this area.
Comments on the Quality of English LanguageOverall the used language requires grammatical revision as there are multiple ungrammatical sentences as well as punctuation issues, such as line 55 "According to the classic stress and coping theory from Lazarus and Folkman (1984), which has long emphasized the importance of coping in the relationship between personality and psycho-adjustment, it’s reasonable to support that there are a variety of ways optimism can be correlated with SWB, especially through the routines of stress coping styles, such as self-control strategies, cognitive reappraisal, emotional regulations, social support, and the like"
in line 103, "The second type involves longitudinal studies, demonstrating that optimism is not only linked to lower depression symptoms initially but also predicts lower depression and higher life satisfaction at subsequent time-points, even after accounting for the initial levels"
Author Response
Dear reviewer,
I made some new corrections and please see the attachment.

Reviewer 2 Report
Comments and Suggestions for Authors
This study primarily investigates the relationships among the concepts of optimism, coping, and Subjective Well-Being (SWB) in Chinese adults, with the aim of extending our understanding of how optimism influences human happiness and psychological adjustment from the perspective of positive cognitive constructs. The results from three empirical studies consistently demonstrate that optimism maintains a significant and stable impact on both positive and negative coping styles, as well as SWB and its components. Specifically, optimism is positively associated with positive coping, SWB, and the domains of Life Satisfaction (LS) and Positive Affect (PA). In contrast, it is negatively correlated with negative coping styles and Negative Affect (NA) in SWB. Furthermore, the findings from the current research suggest that positive coping, rather than negative coping, plays a crucial mediating role in the relationship between optimism and SWB. This study makes contributions to both theory and practice. However, the following suggestions are provided for the authors' consideration.
1. Regarding the methods section in the abstract, it is suggested that the author provides more specific details and descriptions.
2. On page 5, please explain the rationale behind the choice of measurement tools used in Study One, and it is also recommended to provide explanations for the selection of measurement tools in Studies Two and Three.
3. For the measurement of optimism, where Cronbach's alpha is 0.66 and does not meet the minimum standard of 0.70, please explain the reasons for this situation.
4. Please supplement information on how the results of this study can be specifically applied in practice.
Author Response

(The authors gave the same response as above.)

Reviewer 3 Report
Comments and Suggestions for Authors
The study based on the relationship between optimsim and SWB is rigorously carried out.
Study 1: It would be good to add the procedure as well (a short phrase regarding the way to administer the survey)
Study 2 and 3: It should be mentioned if the instruments used are validated.
Additional comments:
The relationship between optimism and SWB has been investigated for a long time. What the study brings new is the mediating mechanism of coping in the respective relationship and the fact that the research is carried out in another cultural model, the Chinese one, than in W.E.I.R.D. The study proposes 7 hypotheses built on the basis of previous literature.
Lines 190-192: Measures - to mention whether the 3 questions used are part of the instrument or the three constitute the entire instrument. If they are extracted from an instrument, what is its name?
Lines 270-276: 3.3.1 Testing for common method bias.
For Harman's single-factor test, it is important to use the unrotated factor solution and not the rotated one as it appears in the text. (see also the study of Tehseen et al. (2017) Testing and Controlling for Common Method Variance: A Review of Available Methods, Journal of Management Sciences, 4(2): 146-175, DOI: 10.20547/jms.2014.1704202)
Lines 280-281 - data analysis software (for all three studies) should be moved to the paragraph ”The present research”
The results are discussed in comparison with previous researches. The references are appropriate.
I consider that the full name should be given in the title and not the acronym SWB.
Author Response

(The authors gave the same response as above.)
